# Inhibition of PKCθ Improves Dystrophic Heart Phenotype and Function in a Novel Model of DMD Cardiomyopathy

**DOI:** 10.3390/ijms23042256

**Published:** 2022-02-18

**Authors:** Jacopo Morroni, Leonardo Schirone, Valentina Valenti, Clemens Zwergel, Carles Sánchez Riera, Sergio Valente, Daniele Vecchio, Sonia Schiavon, Rino Ragno, Antonello Mai, Sebastiano Sciarretta, Biliana Lozanoska-Ochser, Marina Bouchè

**Affiliations:** 1Department of Anatomical, Histological, Forensic Medicine and Orthopaedic Sciences, Section of Histology and Embryology, Sapienza University of Rome, 00161 Rome, Italy; jacopo.morroni@uniroma1.it (J.M.); carles.sanchezr@gmail.com (C.S.R.); biliana.lozanoska-ochser@uniroma1.it (B.L.-O.); 2Department of Medical and Surgical Sciences and Biotechnologies, Sapienza University of Rome, 00185 Rome, Italy; leonardo.schirone@uniroma1.it (L.S.); daniele.vecchio@uniroma1.it (D.V.); sonia.schiavon@uniroma1.it (S.S.); sebastiano.sciarretta@uniroma1.it (S.S.); 3Department of Cardiology, Ospedale Santa Maria Goretti, 04100 Latina, Italy; valevale2012@hotmail.com; 4Department of Drug Chemistry and Technologies, Sapienza University of Rome, 00185 Rome, Italy; clemens.zwergel@uniroma1.it (C.Z.); sergio.valente@uniroma1.it (S.V.); rino.ragno@uniroma1.it (R.R.); antonello.mai@uniroma1.it (A.M.); 5Department of AngioCardioNeurology, IRCCS Neuromed, 86077 Pozzilli, Italy

**Keywords:** DMD, cardiomyopathy, PKCθ, inflammation

## Abstract

Chronic cardiac muscle inflammation and subsequent fibrotic tissue deposition are key features in Duchenne Muscular Dystrophy (DMD). The treatment of choice for delaying DMD progression both in skeletal and cardiac muscle are corticosteroids, supporting the notion that chronic inflammation in the heart plays a pivotal role in fibrosis deposition and subsequent cardiac dysfunction. Nevertheless, considering the adverse effects associated with long-term corticosteroid treatments, there is a need for novel anti-inflammatory therapies. In this study, we used our recently described *exercised mdx* (*ex mdx*) mouse model characterised by accelerated heart pathology, and the specific PKCθ inhibitor Compound 20 (C20), to show that inhibition of this kinase leads to a significant reduction in the number of immune cells infiltrating the heart, as well as necrosis and fibrosis. Functionally, C20 treatment also prevented the reduction in left ventricle fractional shortening, which was typically observed in the vehicle-treated *ex mdx* mice. Based on these findings, we propose that PKCθ pharmacological inhibition could be an attractive therapeutic approach to treating dystrophic cardiomyopathy

## 1. Introduction

Duchenne Muscular Dystrophy (DMD) is a severe X-linked genetic uncurable disease that leads to progressive skeletal and cardiac muscle wasting, affecting roughly 1:3000–5000 of males. It is caused by mutations in the *DMD* gene, encoding for the dystrophin protein [1,2,3,4], a long rod-shaped cytoplasmic protein that exerts a fundamental structural role in muscle, stabilising the sarcolemma. These mutations often result in reading-frame shifts and generation of premature stop codons, producing truncated, non-functional dystrophin. The absence of dystrophin leads to fragile fibres, which become injured during muscle contraction. Sarcolemma leaks and breaks can cause an excessive calcium influx in the cells and the release of cytosolic antigens with subsequent innate inflammatory response, often leading to necrotic cell death and fibrotic tissue deposition [1,2,5,6,7]. Both skeletal and cardiac muscle are affected by the lack of dystrophin. The main clinical manifestations of the pathology are progressive muscle weakness and wasting, with patients often wheelchair-bound in their second decade of life [1,2,4,5,6,7]. The diaphragm is particularly damaged so that most of the patients need nocturnal assisted ventilation within their third decade. In the past, respiratory failure was the leading cause of death amongst patients; nowadays, with the development of new assisted ventilation techniques and palliative treatments, and the subsequent increase of patients’ lifespan, cardiac failure is emerging as the main cause of death [6,7,8,9,10]. DMD cardiomyopathy is characterised by early fibrosis, hypertrophy, and subsequent left ventricle dilation, as well as sustained inflammation [5,9]. Cardiomyopathy develops early in DMD patients: more than 60% of 10-year-old males present clear signs of cardiomyopathy before the loss of ambulation, which increases up to 90% among 18-year-old patients [6,8,9,10,11,12,13].

To date, no definitive cure exists for DMD. Several therapeutic approaches are being tested for their ability to restore dystrophin synthesis: viral vectors carrying micro-dystrophin isoforms, exon skipping compound or stop codon readthrough agents, as well as gene editing with the novel CRISPR/Cas technology [14,15,16,17,18]. Nevertheless, until we succeed in restoring dystrophin production, novel palliative treatments will continue to be investigated in order to slow down disease progression and improve patients’ quality of life [9,10,18]. At present, the golden standard for treating DMD is a life-long administration of corticosteroids [6,7,9,10,11,18]. Corticosteroids are effective in promoting muscle maintenance and, more importantly, in delaying both losses of ambulation and cardiomyopathy progression. Their beneficial effects are primarily due to their immunosuppressive action, supporting the notion that the immune system has a pivotal role in disease progression [19,20,21,22]. Indeed, several studies have shown that depletion of immune populations such as T cells and neutrophils, as well as genetic ablation or inhibition of inflammatory cytokines, ameliorated muscle phenotype and function in the murine model of DMD, the *mdx* mouse [23,24,25,26,27,28,29]. However, given the many side effects associated with the use of corticosteroids, there is an urgent need for alternative approaches to target the inflammatory component of this disease.

Previously we demonstrated the potential of using C20, a PKCθ inhibitor, to ameliorate skeletal muscle and diaphragm pathology by targeting early T cell recruitment [30,31]. PKCθ is highly expressed in T lymphocytes, where it plays a crucial role in their activation by participating in the stable formation of the immunological synapse and the amplification of the TCR-mediated signals, thus representing an attractive target for anti-inflammatory interventions [32,33,34,35,36,37,38,39]. Our group previously demonstrated that both genetic ablation and pharmacological inhibition of PKCθ, using the highly specific inhibitor Compound 20 (C20) [30,31,40], reduced inflammation, necrosis, and fibrosis in skeletal muscle of *mdx* mice, together with an increase in muscle performance and preservation of the muscle stem cell (MuSCs) pool [41,42]. We, therefore, wondered whether this therapeutic approach could also be used to ameliorate DMD cardiomyopathy. Since the commonly used *mdx* mouse model presents a much milder cardiac pathology that develops late in the mice lifespan [43,44,45], we used our recently described “*exercised mdx*” (*ex mdx*) mouse model, which is characterised by accelerated heart pathology, with clear signs of cardiomyopathy, such as diffuse ventricular fibrosis, appearing as early as 12 weeks of age [46].

We showed that C20 treatment reduced the number of all the immune cell components infiltrating the dystrophic heart, suggesting a critical role of immune cells in DMD-related cardiac inflammation. The hearts of *ex mdx* treated with C20 showed a strong reduction in cardiomyocyte necrotic cell death, as well as a sharp decrease of the ventricular fibrotic area. Heart function was also improved by the treatment, with C20-treated mice showing increased left ventricle fractional shortening. Overall, these results suggested that PKCθ inhibition may be considered as an attractive novel therapeutic approach to ameliorate the phenotype and function of dystrophic cardiac muscle. 

## 2. Results

### 2.1. C20 Treatment Reduces Immune Cell Infiltration in Ex Mdx Heart 

To assess whether C20 could ameliorate dystrophic heart phenotype and function, we used the exercised *mdx* model (*ex-mdx*) we previously described and characterised [46]. As summarised in Figure 1A, *mdx* mice were divided into three groups at 3 weeks of age. One group received C20 treatment, while another one received vehicle. A third group of non-exercised, non-treated, age-matched *mdx* were used as control. Vehicle and C20-treated groups were exercised as previously described [46], starting at 4 weeks of age, with two 1-h long sessions per week, at the speed of 20 cm/s, for 8 weeks. Since in the present study the C20 treatment needed to be administered for the duration of 9 weeks, a much longer duration than in our previous works, we first tested the efficacy of daily (as we used before) versus twice-a-week administration of C20 (5 mg/kg) in lowering the number of circulating CD3^+^ and CD11b^+^ immune cells. After two weeks of treatment, we found that twice-weekly administration was as effective as daily administration (Appendix A). C20 or vehicle were thus administered to mdx mice from 3 weeks of age (one week before starting the exercise session) via intraperitoneal injection, twice a week at 5 mg/kg [30,31]. At the end of the exercise protocol, at the age of 12 weeks, mice from both groups, together with unexercised age-matched controls, were sacrificed, and hearts were harvested for analysis. Interestingly, during the exercise sessions, we observed a decrease in exhaustion among the C20-treated mice compared with vehicle-treated (Figure 1B), suggesting that treatment with the established dose and frequency of C20 might be effective in ameliorating running performance. 

### 2.2. C20 Treatment Reduces Immune Cells Infiltration of the Heart

Next, considering that PKCθ is required to mount an effective immune response, we investigated the effect of its inhibition on immune cell infiltration in the dystrophic heart by cytofluorimetric analysis. Our complete gating strategy to identify different immune cells is shown in Appendix A. The number of total CD45^+^ hemopoietic cells was significantly reduced in the heart of C20 treated mice compared with vehicle-treated counterparts (Figure 2A,B). The abundance of CD45^+^ cells observed in C20 treated *ex mdx* was similar, or even lower, compared to unexercised age-matched *mdx* (Figure 2B). All the immune cell populations examined were significantly reduced in number, except for neutrophils, which, although reduced, did not reach statistical significance (Figure 2C–G). These results demonstrated that inhibiting PKCθ-dependent T-cell infiltration reduces inflammation in the dystrophic heart.

### 2.3. C20 Treatment Reduces the Cells Infiltration Area and Cardiomyocyte Necrosis in Mdx Ventricles

To examine heart muscle organisation in C20 and vehicle-treated *ex mdx*, ventricle cryosections were stained with haematoxylin and eosin (H&E). As shown in Figure 3A, we found no apparent alterations in cardiac muscle tissue organisation; however, in line with the cytofluorimetric results, we found a reduction in the cell infiltration total area in C20 treated hearts compared with controls (Figure 3B), together with a decrease in the size of the infiltrating cell patches (Figure 3C), while no change was found in their number, normalised per square millimetre (Figure 3D). 

Since immune cell infiltration might be triggered by necrotic cell death of cardiomyocytes [47,48], we analysed ventricular cardiomyocyte necrosis in C20 or vehicle-treated *ex mdx* by immune-staining of the ventricle cryosections with anti-mouse IgG conjugated to a TRITC fluorochrome. As shown in Figure 4A,B, the necrotic ventricle area was significantly reduced in the C20 treated mice compared with vehicle-treated controls, with levels comparable to unexercised age-matched *mdx*. Taken together, these results demonstrated that C20 treatment ameliorates dystrophic heart phenotype in the *ex mdx* model by reducing immune cell infiltration and cardiomyocyte necrosis.

### 2.4. C20 Treatment Reduces Fibrotic Tissue Deposition in Ex-Mdx Ventricles

Excessive cardiomyocyte necrotic cell death and subsequent cell infiltration lead to fibrotic tissue deposition in heart ventricles [48,49]. To examine the effect of C20 treatment on heart fibrosis in the *ex mdx* mice, cryosections of the heart ventricles were stained with Sirius red, and the extent of collagen deposition was analysed using ImageJ software. As shown in Figure 5A and B, C20 treatment resulted in a strong decrease of the ventricular fibrotic area compared with vehicle-treated controls. Moreover, the level of ventricle fibrosis in the C20 treated *ex mdx* was similar to the level observed in unexercised age-matched *mdx*. Accordingly, Collagen1α mRNA was significantly decreased in C20-treated hearts compared with vehicle-treated controls, as shown by qRT-PCR (Figure 5C), accompanied by a decrease in the pro-inflammatory cytokine IL-6 transcript (Figure 5D). Taken together, these results showed that C20 treatment reduces fibrotic tissue deposition in the dystrophic heart.

### 2.5. Heart Function Is Preserved in C20-Treated Exercised Mdx

In DMD-related cardiomyopathy, fibrotic tissue deposition is associated with ventricular hypertrophy and is followed by progressive left ventricle dysfunction, reduced heart contractility, and ejection volume [5,6,9,10]. Since we found a reduction in the ventricular fibrotic area in C20-treated *ex mdx*, we wondered whether PKCθ inhibition can ameliorate heart function. We, therefore, performed an echocardiographic analysis in C20 and vehicle-treated mice; age-matched unexercised *mdx* mice were added as a baseline control. The main parameters investigated are summarised in Table 1.

As shown in Table 1 and in Figure 6A,B, C20 treatment prevented the reduction of left ventricle fractional shortening observed in vehicle-treated *ex mdx*. No alterations were seen in wall or septum thickness, or in chamber dilation as well as in heart rate. These results demonstrated that C20 treatment can preserve cardiac function in the *ex mdx* hearts.

## 3. Discussion

This study demonstrated that blunting dystrophic heart inflammation through PKCθ inhibition can ameliorate heart phenotype and improve heart function in a novel model of DMD-related cardiac pathology. 

We used the *exercised mdx* mice we recently characterised in order to test pharmacological approaches, because they display an accelerated and worsened dystrophic heart phenotype [46]. In the *mdx* mouse model in fact, the cardiac workload induced by the exercise is not tolerated because of the absence of dystrophin. By contrast, it is well known that chronic, moderate-intensity exercise on healthy mice does not induce any detrimental effect on cardiac phenotype and function, but beneficial instead [50,51,52,53,54,55,56,57]. We therefore avoided exercising wild-type mice, and we compared the cardiac phenotype of the *exercised mdx* mice treated or not with the inhibitor to age-matching *mdx* mice when no signs of heart abnormalities were evident without exercise. Importantly, we previously showed that C20 treatment did not have detrimental effects on cardiac phenotype of healthy mice in terms of tissue organization and gene expression, making it a promising pharmacological strategy [31]. As discussed before, PKCθ represents an attractive target for anti-inflammatory interventions, since, given its crucial role in TCR-mediated signals, its inhibition might result in immune-modulation rather than a generic immune-suppression [33,36,58,59,60].

We used this approach since it is now well established that the immune response plays a critical role in DMD pathology progression [28,61,62], and interfering with the onset or the amplification of the immune response can ameliorate dystrophic muscle phenotype and/or function [23,24,25,26,27,29,63] as well as heart pathology [64,65,66,67]. It is well established that T cells orchestrate and amplify the immune response by secreting inflammatory cytokines and by recruiting other immune cells at the site of inflammation through various chemokines release [26,68,69], and targeting T cells may represent a therapeutic strategy for diseases that share chronic inflammation as a common feature. PKCθ raised attention as an attractive protein target to interfere with T cells activity because of its crucial role in T cells activation and proliferation [33,36,38,39]. We previously showed that the genetic ablation of PKCθ in the *mdx* resulted in a striking improvement of dystrophic skeletal muscle phenotype and function as well as a preserved MuScs pool [42,70]. Interestingly, ablating PKCθ in skeletal muscle but not in hematopoietic cells could not rescue the dystrophic muscle phenotype [70]. Moreover, the pharmacological inhibition of PKCθ, with the specific inhibitor C20, counteracted skeletal muscle dystrophic progression [30,31,41]. In particular, we showed that among the immune cell populations, T lymphocytes are the main target of C20 treatment, and targeting their early recruitment to dystrophic muscle reduced the number of all the other infiltrating immune populations, such as inflammatory monocytes or macrophages [30].

Similarly, in this study, we show that C20 treatment strongly reduces the number of all the immune cell types infiltrating the exercised dystrophic heart, as well. Indeed, not only T cells, but all the myeloid cell populations examined, such as recently recruited monocytes, macrophages, and neutrophils were reduced in the heart of C20-treated exercised *mdx.* Considering that in the dystrophic heart environment, ventricular damage could be amplified by the immune response [9], and on the basis of previous results we obtained on the skeletal muscle [30,31], we propose that PKCθ inhibition by C20 administration might blunt the immune response sustained by T cells recruitment, dampening the amplification of the damage and the subsequent cardiomyocytes necrosis and fibrotic tissue deposition (Figure 7).

In this study, we showed that C20 treatment fully prevented the dystrophic cardiac pathology observed in 12-week-old exercised mdx mice. Indeed, the level of fibrosis, inflammation, and necrosis was similar to the control age-matched 12-week-old *mdx,* in which no significant cardiac abnormalities are yet detectable. In line with the histological results, we found that C20 treatment prevented the increase in collagen 1α1 transcription and in IL-6 mRNA observed in exercised mice, bringing it back to the level in control animals. Interestingly, IL-6 blockade or depletion was shown to ameliorate heart fibrosis in other cardiopathic mice models [71,72]. Indeed, the inflammation-necrosis-fibrosis axis has been described as pivotal in driving heart failure [47,48,73]. In our study, C20 treatment prevented the decrease in left ventricle fractional shortening observed in the vehicle-treated *ex mdx*, in line with the decrease in ventricular fibrotic tissue deposition. It is worth mentioning that C20 treatment also reduced the frequency of treadmill exhaustion during the exercise protocol. This fact could be attributed mainly to the preserved heart function since, as we described previously, the lower limb and diaphragm muscle are only slightly affected by this exercise protocol [46].

Nevertheless, we cannot rule out that other PKCθ-expressing cells, besides immune cells, might be targeted by C20 and contribute to the ameliorated dystrophic heart phenotype and function. PKCθ has been suggested to be involved in other biological processes, such as platelet activation, insulin response, or cancer progression. Nevertheless, the role of this protein in other cell types appears not to be crucial [74,75,76]. By contrast, the essential role of PKCθ in T cells is well established and characterised [32,34,35,37], making PKCθ an attractive target for anti-inflammatory approaches [33,36,38,60,77].

Considering that PKCθ is also expressed in murine cardiomyocytes, the possibility that long-term treatment with C20 might interfere with cardiomyocytes function should be considered. Indeed, although PKCθ^−/−^ mice develop normally and are healthy [42,70], we previously showed that lack of PKCθ increased heart fibrosis over time and made cardiomyocyte more prone to workload-induced apoptosis [78]. However, in that model, PKCθ is lacking constitutively from embryonic development throughout the entire life, a condition not mirrored by administering a PKCθ inhibitor twice a week at a relatively low dose. As previously mentioned, C20 administration did not have significant effects on healthy, non-inflamed hearts [31]. In any case, in a chronic inflammatory environment, as in the dystrophic heart, the beneficial effects of blunting the immune response might overcome the eventual detrimental effects on cardiomyocytes, similarly to what we found in dystrophic skeletal muscle [30,31]. Finally, since PKCθ is not expressed in the human heart [79], but it is in human immune cells, it may represent an attractive target to ameliorate the human dystrophic heart pathology.

In conclusion, we showed that inhibition of PKCθ reduced dystrophic heart inflammation, necrosis, and fibrosis, and ameliorated heart function in a model of worsened DMD cardiomyopathy. We believe that this could represent an attractive palliative approach to slow down the progression of dystrophic cardiomyopathy.

## 4. Materials and Methods

### 4.1. Animal Models

C57BL/10ScSn-*Dmd^mdx^* mice were purchased from the Jackson Laboratory (Bar Harbor, ME, USA). The mice were housed in the Histology Department-accredited animal facility at the University of Sapienza. All the procedures were approved by the Italian Ministry for Health and were conducted according to the EU regulations and the Italian Law on Animal Research. Only males were used.

### 4.2. Treadmill Exercise

The exercise was carried out as previously described [46] using a five-lane motorised treadmill (LE 8710, PanLab S.L.U., Barcelona, Spain) supplied with shocker plates. Four-week-old mdx mice were exercised twice a week for 8 weeks. At the end of the exercise protocol, at 12 weeks of age, mice were sacrificed and processed for analyses. To set up the exercise protocol, we followed the general indication contained in the S.O.P. DMD_M_2.1.001 by De Luca et al., 2008. After 10 min of acclimation on the stationary treadmill, the exercise started with a 10 min-long “warm-up” session in order not to stress or exhaust the mice too fast. Warm-up started at a speed of 10 cm/s, increasing 2 cm/s every 2 min. Then, 50 min of exercise was performed at a constant speed of 20 cm/s. We recommend spacing out two sessions of 25 min of exercise (20 cm/s) with a 5 min-long pause. A critical adjustment we followed was not to use electric shocker plates to induce the mice to the run but to use gentle manipulation or a physical obstacle. This adjustment reduced stress in mice and sensibly increased their ability to complete the exercise session, avoiding exhaustion.

### 4.3. C20 Treatment

C20 was prepared as a mother solution at the concentration of 100 ug/uL in DMSO and then diluted 1:100 in 0.9% NaCl saline solution. The vehicle was prepared diluting DMSO 1:100 in the same saline solution. Three-week-old *mdx* mice were intraperitoneally injected twice a week with C20, at the dose of 5 mg/kg, or with vehicle, in a total volume of around 0.1 mL, using 500 μL syringes with 27 G needles. The treatment was maintained throughout the exercise protocol, which started one week later at the age of 4 weeks, for the following 8 weeks. The bi-weekly C20 or vehicle injection preceded the bi-weekly exercise session of about 24 h. Mice were sacrificed 48 h after the last exercise session.

### 4.4. Histology

Heart ventricles were embedded in tissue-freezing medium after dissection (O.C.T. Compound, Sakura 4583, Japan) and snap-frozen in liquid nitrogen-cooled isopentane. Frozen heart ventricles were cut into 8 μm sections (four sections per heart) and stored at −20 °C until use. Histochemistry and immunofluorescence analyses were performed as previously described [30,31,42]. 

Briefly, for histological analysis, the sections were stained with haematoxylin/eosin or with Sirius red/picric acid (both from Sigma-Aldrich, Saint Louis, MO, USA). The sections were photographed in a Zeiss Axioskop 2 Plusfluorescence microscope, using a 10× objective and a 10× eyepiece. The whole ventricular surface was acquired for each section. Quantification of cell infiltration patches size and number, as well as total ventricular area, was performed using built-in functions in ImageJ open-source software (https://imagej.nih.gov/ij/, accessed on 31 December 2021). Quantification of collagen deposition was determined using the Color Deconvolution plugin (by G. Landini, https://imagej.net/plugins/colour-deconvolution, accessed on 31 December 2021) of ImageJ software. 

For immunostaining of the necrotic areas, permeabilization in methanol (6 min at −20 °C) was performed on cryosections after fixation. After three PBS washes, sections were blocked in 5% goat serum and incubated with rabbit anti-laminin primary antibody (Life Technologies, Carlsbad, CA, USA) for 1 h at room temperature, at the dilution of 1:400 in 1% PBS-BSA. After three washes in 1% PBS-BSA, sections were incubated with goat anti-mouse IgG antibody coupled to TRITC and goat anti-rabbit IgG coupled to Alexa Fluor 488 (both from Life Technologies, Carlsbad, CA, USA). Nuclei were counterstained with Hoechst 33,342 (Fluka, Charlotte, NC, USA). The sections were photographed in a Zeiss Axioskop 2 Plusfluorescence microscope, using a 10× objective and a 10× eyepiece. All the ventricular necrotic areas were acquired for each section. The images were then analysed using ImageJ software. 

### 4.5. Flow Cytometry

Cytofluorimetric analysis was performed as previously described [29,30]. Briefly, hearts were collected and cut into small pieces with a blade, and then incubated with collagenase type IV for 1 h and 30′ at 37 °C with agitation. The obtained cell suspension was passed through a 70 μm and then a 40 μm cell strainer; the cells were then counted on a hemacytometer, collected by centrifugation at 1200 rpm, and suspended in 200 μL of calcium/magnesium-free PBS (phosphate-buffered saline) with 2% FBS (foetal bovine serum). They were then divided into two tubes for the staining. The cells were then incubated on ice for 30 min with the following antibodies: CD45 PE/Cy7, F4/80 APC, Ly6g PE Fluor 610, CD11b APC/Cy7, CD206 PERCP/Cy5.5 Ly6c BV-510, I-Ab FITC (tube 1) and CD45 PE/Cy7, CD3 PERCP Cy5.5, B220 BV-510, CD4 AF488, and CD8 PE (tube 2), all by Biolegend, (San Diego, CA, USA). Cells were then washed with 3 mL of calcium/magnesium-free PBS and resuspended in 200 μL of calcium/magnesium-free PBS with 2% FBS. Samples were acquired with a CyAn ADP (Agilent DAKO, Santa Clara, CA, USA), and acquired data were analysed using FlowJo software version 10.1.

### 4.6. qRT-PCR

Total RNA from ventricle tissue was extracted using the TRIsure solution (Bioline, London, UK) and converted in cDNA using the SensiFast cDNA Synthesis kit from Bioline, according to the supplier’s instructions. PCR amplification was performed using the SensiMix SYBR Lo-Rox Mix, from Bioline, following the manufacturer’s protocol. All PCR reactions were carried out in duplicate. All qPCR results are expressed as relative ratios of the target cDNA transcripts to GAPDH and normalised to those of the reference condition. The reaction was carried out using a 7500 Real-Time PCR System, and the analysis was carried out with the 7500-software, both from Applied Biosystem, Waltham, MA, USA. To amplify the genes of interest, we used the following primers pairs: 

GAPDH 

(for) 5′-ACCCAGAAGACTGTGGATGG-3′ 

(rev) 5′-CACATTGGGGGTAGGAACAC-3′: 

Collagen1α1 

(for) 5′-ACCCAGAAGACTGTGGATGG-3′ 

(rev) 5′-CAGATTGGGGGTAGGAACAC-3′, 

IL-6

(for) 5′-CCCGAAGCGGACTACTATGC-3 

(rev) 5′-CATAGATGGCGTTGTTGCGG-3′.

### 4.7. Echocardiography

Echocardiographic analyses were performed as previously described [80,81]. Briefly, mice were anaesthetised with 2.5% avertin at the dose of 250 mg/kg (Sigma, Saint Louis, MO, USA, T48402) and mice chests were shaved. All left measurements were taken in M-mode short-axis using a VEVO 3100 (Visualsonics, Toronto, ON, Canada) with an mx550d probe.

### 4.8. C20 Synthesis

C20 was synthesised at the Dept. of Drug Chemistry and Technologies, La Sapienza, Roma, according to Cywin et al., 2007 [40]. Compound I (0.62 mmol) was dissolved in AcOH at 0 °C and then KSCN (0.65 mmol) was added portion wise. Upon completion of the reaction (TLC: EtOAc:Hex 2:1, 2 h), the mixture was quenched with water and the precipitate was filtered off. The so obtained compound II (2-chloro-5-nitro-4-thiocyanatopyrimidine) was washed with diethylether and used directly in the next step. m.p.: 115–117 °C; yield: 60%; ^1^H-NMR (CDCl_3_, 400 MHz) *δ* 9.40 (1H, s). 

Compound II (0.46 mmol) was suspended in abs EtOH, then *o*-trifluoromethoxybenzylamine (0.46 mmol) was added followed by the dropwise addition of triethylamine (0.92 mmol). The resulting precipitate was filtered off after 16 h of stirring at room temperature (TLC: EtOAc:Hex 2:1) providing the desired intermediate III (5-nitro-4-thiocyanato-N-(2-(trifluoromethoxy)benzyl)pyrimidin-2-amine), which was used without further purification in the next step. m.p.: 190–192 °C; yield: 51%; ^1^H-NMR (DMSO, 400 MHz) *δ* 4.79 (2H, dd), 7.36–7.54 (4H, m), 9.51 (1H, s), 9.75 (1H, m).

Compound III (0.39 mmol) was suspended in dry DCM and 1,4-Cyclohexanebis(methylamine) (1.56 mmol) and the reaction was left stirring for 16 h at rt. The crude was loaded on a silica gel column using chloroforom/methanol/ammonia 12:1:0.1 as the eluent system, yielding pure C20 (N4-((4-(aminomethyl)cyclohexyl)methyl)-5-nitro-N2-(2-(trifluoromethoxy) benzyl)pyrimidine-2,4-diamine). m.p.: 105–107 °C; yield: 52%; ^1^H-NMR (DMSO, 400 MHz) *δ* 0.50–1.50 (13H, m), 2.86 (2H, d), 3.21 (1H, t), 4.63 (2H, dd), 7.34–7.41 (4H, m), 8.69 (1H, m), 8.71 (1H, bs), 8.89 (1H, s); MS (M + H) = 455.01.

### 4.9. Statistical Analysis

All statistical analyses were performed using Prism software, version 6 (GraphPad Software, Inc., La Jolla, CA, USA). Data are presented as mean ± SEM. Unpaired two-tailed Student’s *t*-test with Welch’s correction was used for statistical comparison between two groups, and one-way ANOVA (with Bonferroni’s correction for multiple comparisons) was used for comparisons between multiple groups. A *p*-value of ≤0.05 was considered statistically significant.

## Figures and Tables

**Figure 1 ijms-23-02256-f001:**
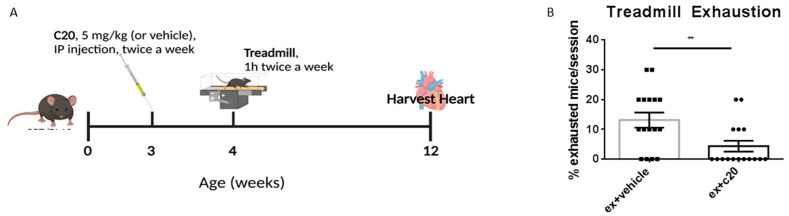
C20 twice-a-week administration increases running performance in the *mdx* mice. (**A**) A schematic description of the C20 treatment and exercise protocol. (**B**) Percentage of mice exhausted per exercise session, expressed as mean +/− S.E.M.; ** *p* < 0.01, unpaired *t*-test w/Welch’s correction. N = 10 mice/exercise session.

**Figure 2 ijms-23-02256-f002:**
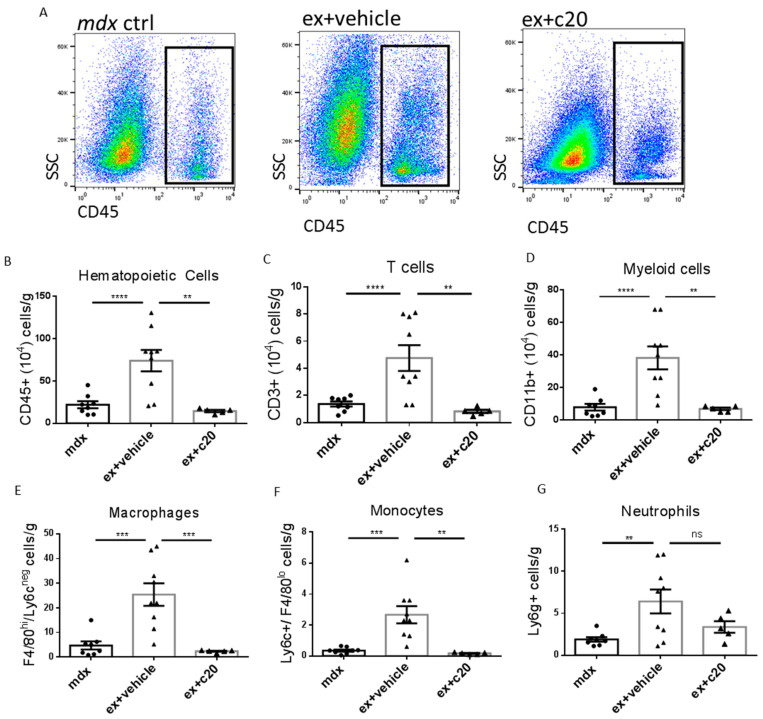
Reduced immune infiltration in C20-treated exercised *mdx* hearts compared with controls. (**A**) Representative images showing the gating for CD45^+^ heart immune infiltrating cells in control *mdx*, vehicle-treated, and C20-treated exercised *mdx*. (**B**–**G**) Quantification of total CD45^+^ cells (**B**) CD3^+^ T cells (**C**), CD11b^+^ myeloid cells (**D**), F4/80^hi^ macrophages (**E**), Ly6c^hi^-F4/80^−^ inflammatory monocytes (**F**), and Ly6g^+^ neutrophils (**G**) in control *mdx*, vehicle-treated, and C20-treated exercised *mdx* hearts, expressed as number of cells normalised per gram of tissue. Data are shown as media +/− S.E.M. *n* = 5–9 independent samples per group ** *p* < 0.01, *** *p* < 0.001, **** *p* < 0.0001, ordinary one-way ANOVA with Bonferroni correction for multiple comparisons.

**Figure 3 ijms-23-02256-f003:**
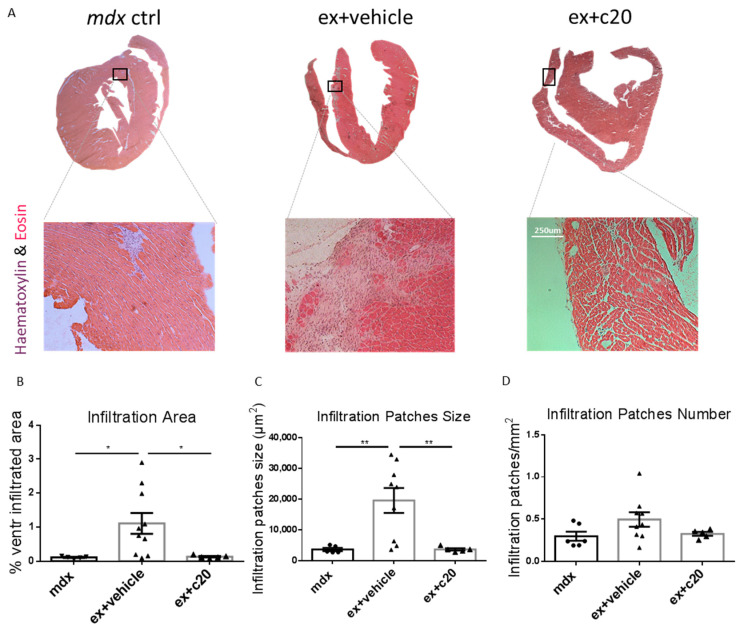
Decreased cell infiltration area in C20-treated exercised *mdx*. (**A**) Representative H&E staining showing myocardium morphology and the infiltrating cell patches in control *mdx* and vehicle or C20-treated exercised *mdx*. (**B**) Quantification of cell infiltration area expressed as percentage over the total ventricular area, as determined in H&E stained cryosections. (**C**) Quantification of the size of cells infiltration patches expressed in µm^2^. (**D**) Quantification of the number of cells infiltration patches, normalised per mm^2^. Data from (**B**,**D**) are expressed as mean +/− S.E.M.; *n* = 5, 9, five independent samples per group. All quantifications were calculated using ImageJ software, * *p* < 0.05, ** *p* < 0.01, ordinary one-way ANOVA with Bonferroni correction for multiple comparisons.

**Figure 4 ijms-23-02256-f004:**
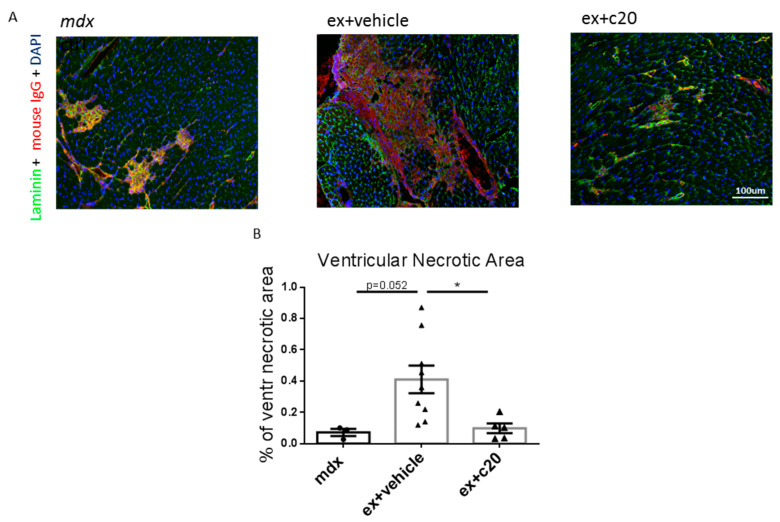
Decreased cardiomyocytes necrosis in C20-treated exercised *mdx* ventricles. (**A**) Representative images showing ventricular necrosis in control *mdx* and vehicle or C20-treated exercised *mdx*, as detected by anti-mouse IgG staining. (**B**) Quantification of the ventricular necrotic area expressed as percentage over the total ventricular area, calculated using ImageJ software. Data are shown as mean +/− S.E.M, *n* = 3, 7, 5 independent samples per group. * *p* < 0.05, ordinary one-way ANOVA with Bonferroni correction for multiple comparisons.

**Figure 5 ijms-23-02256-f005:**
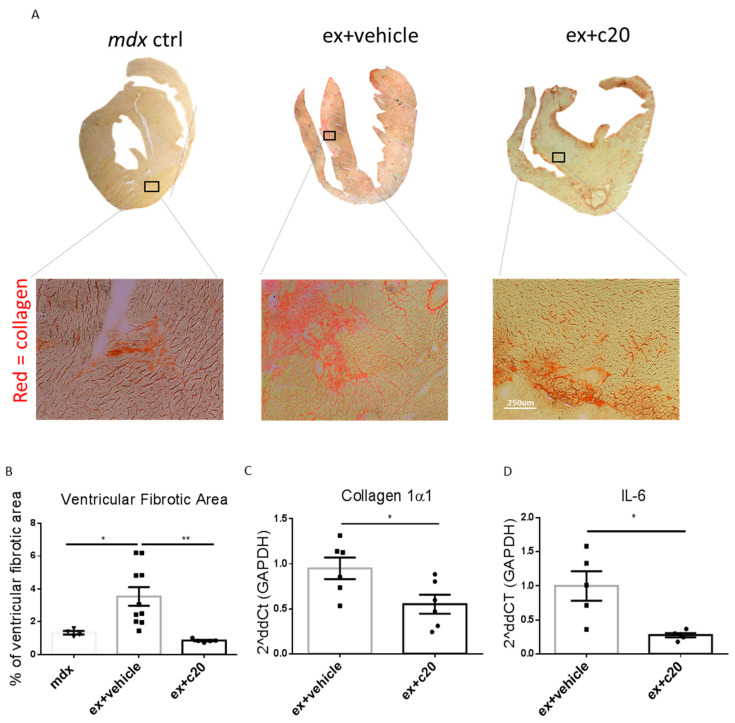
Decreased fibrotic tissue deposition in C20-treated exercised *mdx* ventricles. (**A**) Representative images of ventricular fibrosis in the hearts of control *mdx* vehicle-treated or C20-treated exercised *mdx* mice, evidenced by Sirius red collagen staining. (**B**) Quantification often ventricular fibrotic area over the total ventricular area expressed as percentage and calculated by ImageJ Colour Deconvolution plugin. *n* = 5, 10, 5 independent samples per group. * *p <* 0.05, ** *p* < 0.01, ordinary one-way ANOVA with Bonferroni correction for multiple comparisons. (**C**,**D**) qRT-PCR on total heart RNA for Collagen1α (*n* = 6 independent samples per group) and IL-6 mRNA (*n* = 5 independent samples per group). * *p* < 0.05, unpaired t-test w/Welch’s correction. Data from (**B**,**D**) are shown as mean +/− S.E.M.

**Figure 6 ijms-23-02256-f006:**
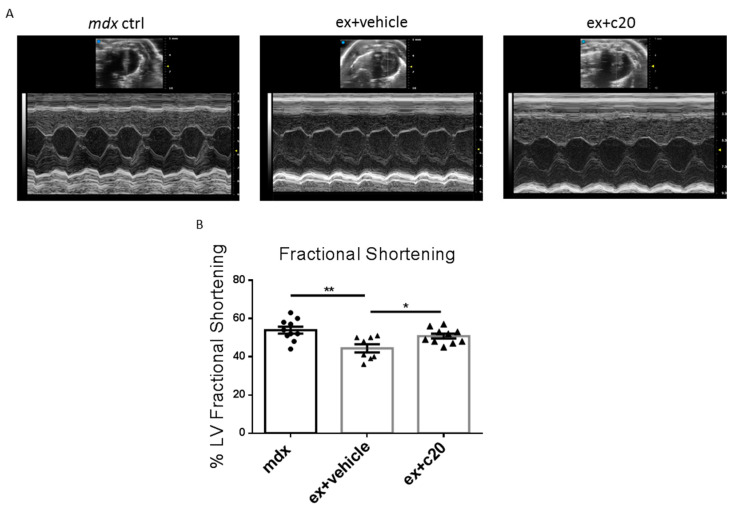
Prevented left ventricular loss of function in C20 treated *ex mdx*. (**A**) Representative images of the echocardiographic analysis of control *mdx* and vehicle or C20-treated *ex mdx*. (**B**) Left ventricle fractional shortening expressed in percentage and calculated on the short axe. *n* = 10, 8, 10 independent samples per group. * *p* < 0.05, ** *p* < 0.01, one-way ordinary ANOVA with Bonferroni correction for multiple comparisons.

**Figure 7 ijms-23-02256-f007:**
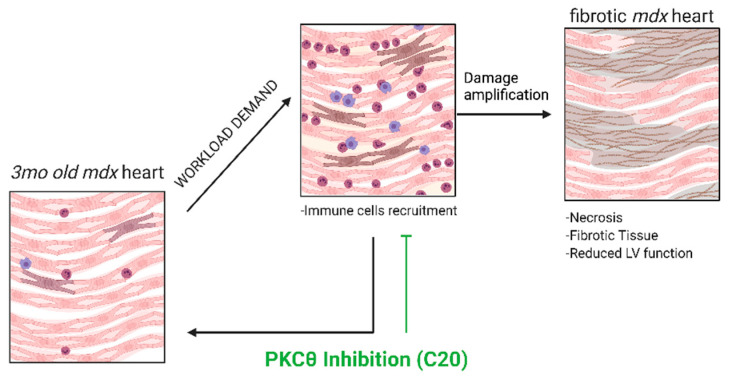
Proposed mechanism of C20 effect. The loss of functional dystrophin causes cardiomyocytes sarcolemma leaks in response to cardiac workload (mimicked by the exercise protocol), promoting immune cells infiltration. Immune cell infiltration amplifies cardiac damage, leading to exacerbated fibrotic tissue deposition. C20 treatment, inhibiting T cells activity, blunts the inflammation in response to CMCs cell death, leading to a decrease in myeloid infiltration and protecting the tissue by the amplification of the damage.

**Table 1 ijms-23-02256-t001:** Summary of the main echocardiographic parameters examined in exercised and control *mdx*. Data are expressed as mean +/− S.E.M., *n* = 8–10/condition ^#^
*p* < 0.05 (*ex mdx* vs. ex + c20), *** p* < 0.01 (*mdx* vs. *ex mdx*), one-way ordinary ANOVA with Bonferroni correction for multiple comparisons.

	End-Diastolic Diameter [mm]	End-Systolic Diameter [mm]	Anterior Wall Thickness [mm]	Posterior Wall Thickness [mm]	Fractional Shortening [%]	Heart Rate (bpm)	Heart Weight/Body Weight [mg/g]
** *mdx* **	2.78 ± 0.044	1.41 ± 0.09	1.11 ± 0.06	1.08 ± 0.03	53.9 ± 1.81	417 ± 16.4	5.95 ± 0.16
**ex+vehicle**	3.04 ± 0.11	1.5 ± 0.1	1.06 ± 0.04	1.16 ± 0.02	44.4 ± 2.12 **	441 ± 32.7	5.53 ± 0.28
**ex+c20**	3.1 ± 0.12	1.53 ± 0.1	1.11 ± 0.07	1.11 ± 0.03	50.8 ± 1.26 ^#^	443 ± 43.8	5.55 ± 0.34

## Data Availability

No new data were created or analyzed in this study. Data sharing is not applicable to this article.

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
