# Peer review of "Inhibition of PKCθ Improves Dystrophic Heart Phenotype and Function in a Novel Model of DMD Cardiomyopathy"

_ijms, 2022, doi:10.3390/ijms23042256_

Round 1

Reviewer 1 Report

Looking for a new solution in therapy of chronic cardiac muscle inflammation the authors decided to determine effects of the Protein Kinase C Theta (PKCΘ) inhibitor - Compound 20 on cardiac muscle fibrosis and cardiac ventricle efficacy.  They used exercised mdx mouse murine model of Duchenne Muscle Dystrophy (DMD). They found that application of C20 improved left ventricle fractional shortening and reduced cardiomyocyte death and fibrotic damage of the heart. Using echocardiographic analysis they found improvement of left ventricle fractional shortening. The authors postulate that inhibition of PKCΘ should be considered as one of the pharmacological methods of treatment of dystrophic cardiomyopaties. 

The study provides interesting results, however they would be much more valid if apart from Mdx, and ExMdx groups treated with C20 or vehicle the control groups treated vehicle and C20 would be included.  Such groups should be included or the authors should convincingly explain why they were not designed. 

Some essential studies determining properties and effects of PKCΘ are not included in the References (ex. Zhang et al; Adv. Pharmacol. 2013).

Author Response

Looking for a new solution in therapy of chronic cardiac muscle inflammation the authors decided to determine effects of the Protein Kinase C Theta (PKCΘ) inhibitor - Compound 20 on cardiac muscle fibrosis and cardiac ventricle efficacy.  They used exercised mdx mouse murine model of Duchenne Muscle Dystrophy (DMD). They found that application of C20 improved left ventricle fractional shortening and reduced cardiomyocyte death and fibrotic damage of the heart. Using echocardiographic analysis they found improvement of left ventricle fractional shortening. The authors postulate that inhibition of PKCΘ should be considered as one of the pharmacological methods of treatment of dystrophic cardiomyopaties. 

The study provides interesting results, however they would be much more valid if apart from Mdx, and ExMdx groups treated with C20 or vehicle the control groups treated vehicle and C20 would be included.  Such groups should be included or the authors should convincingly explain why they were not designed. 

  • We thank the reviewer for raising a proper issue. Indeed, we previously reported that C20 treatment in WT mice did not alter healthy hearts (Marrocco et al., 2017, ref 31), in terms of heart tissue organization and fibrosis (H&E and Masson’s staining) as well as expression level of specific factors such as ANP, TGF-beta or beta MyHC mRNAs. It is widely reported that moderate “endurance” exercise does not exert any detrimental effect in healthy mice hearts, but beneficial. In the present study, we use exercise to worsen and accelerate the dystrophic heart phenotype, as a model to test pharmacological approaches. Considering the “3Rs” recommendation in animal experimental procedures, we think that treating exercised healthy mice would be superfluous and beyond the aim of the study.

Some essential studies determining properties and effects of PKCΘ are not included in the References (ex. Zhang et al; Adv. Pharmacol. 2013).

  • We thank the reviewer for pointing out this omission. We added Zhang, 2013 and Hage-Sleiman, 2015.

Reviewer 2 Report

The results of this manuscript demonstrate that administration of C20, a PKCθ inhibitor, resulted in reduced dystrophic heart inflammation, reduced necrosis and fibrosis, and improvement of heart functionality in a mouse model of DMD-related cardiomyopathy. The manuscript is well written, with appropriate experimental design.  The data are appropriately described and the findings are interesting with a potential as a future therapeutic intervention for the treatment of dystrophic cardiomyopathy.

However, mechanistically the above conclusion is not well established. There is no experimental evidence for a direct connection of PKCθ inhibition of immune cells and the reduced immune cell infiltration.

The authors need to address the following

Major comments

  • Although there is a clear improvement of heart phenotype and function after pharmacological inhibition of PKCθ, the study lacks insights regarding the mechanism underlying this phenomenon.
  • According to the authors, the beneficial effect of the PKCθ inhibitor on the dystrophic heart reflects targeting of the detrimental immune response produced by immune cells. Please address the following:
  1. It is logical to assume that other cell types, besides immune cells, express PKCθ and thus the C-20 may have a broader effect not limited to the immune system. The authors need to provide evidence regarding the cell types that contribute to the cardiac phenotype.

Minor comments

  • Line 209 please check spelling and rephrase
  • In Methods section please clarify if treatment and treadmill exercise occur in the same day.

Author Response

The results of this manuscript demonstrate that administration of C20, a PKCθ inhibitor, resulted in reduced dystrophic heart inflammation, reduced necrosis and fibrosis, and improvement of heart functionality in a mouse model of DMD-related cardiomyopathy. The manuscript is well written, with appropriate experimental design.  The data are appropriately described and the findings are interesting with a potential as a future therapeutic intervention for the treatment of dystrophic cardiomyopathy.

We thank the reviewer for appreciating our study

However, mechanistically the above conclusion is not well established. There is no experimental evidence for a direct connection of PKCθ inhibition of immune cells and the reduced immune cell infiltration.

The authors need to address the following

Major comments

  • Although there is a clear improvement of heart phenotype and function after pharmacological inhibition of PKCθ, the study lacks insights regarding the mechanism underlying this phenomenon.
  • According to the authors, the beneficial effect of the PKCθ inhibitor on the dystrophic heart reflects targeting of the detrimental immune response produced by immune cells. Please address the following:
  1. It is logical to assume that other cell types, besides immune cells, express PKCθ and thus the C-20 may have a broader effect not limited to the immune system. The authors need to provide evidence regarding the cell types that contribute to the cardiac phenotype.

  • We thank the reviewer for raising this issue. We agree that we cannot rule out that non-immune cells expressing PKCθ in the heart might be targeted by the C20, contributing to its beneficial effects, and we modified the discussion accordingly. Nevertheless, it is well-established and characterized the role of PKC-theta in T cells activation, and we show here that C20 treatment reduces T cells infiltration together with all the immune cells. We previously showed that inhibiting early PKCtheta-dependent T cell infiltration in dystrophic skeletal muscle, by C20 treatment, ameliorated skeletal muscle dystrophic phenotype.   We thus believe as T lymphocytes inhibition the major mechanism through which C20 exerts its effects, although the eventual contribution of other cell types within the heart cannot be ruled out.

Minor comments

  • Line 209 please check spelling and rephrase

Done.

  • In Methods section please clarify if treatment and treadmill exercise occur in the same day.

We clarified this issue in the Methods section

Reviewer 3 Report

The manuscript “Inhibition of PKC theta improves dystrophic heart phenotype and function in a novel model of DMD cardiomyopathy” describes the effects of a protein kinase C theta isozyme inhibitor C20 on myocardial inflammation in a Dystrophin myopathy mouse model (mdx), a model previously described in references 43 and subsequently– because of limited cardiac abnormality - adapted with exercise loading in reference 44. The study is interesting and it states interesting ideas about T cell infiltrations in DMD as the initiating pathophysiology, recruited by PKC theta activations. The study is a logical sequel of previous studies on skeletal muscle abnormalities in the DMD model by the same group, and the interpretation of results also rely on these studies.

The authors were faced with a problem of delay in development of cardiac abnormalities in the mouse model, with earliest signs at 12 weeks (reference 44), while also aware of the cardiac effects of a PKC theta knock out mouse model, showing maladaptive cardiac changes (fibrosis and upregulation of ECM and hypertrophy genes) at 8 weeks (reference 53). It means that a PCK theta inhibition cannot start at birth/ too soon when developmental changes are still occurring in the myocardium. Therefore (as illustrated in figure 1) in the present study the mice were treated with C20 or its vehicle from 3 weeks of age with the PKC theta inhibitor C20, the exercise was started at 4 weeks of age and the hearts were studied at 12 weeks. An extra comparison was made with the non-exercise mdx model, which should have some signs as cardiac abnormality as explained in reference 44, but serves as a non-treated non-exercising model. To better understand your findings, a WT control mouse or a C20-treated non-exercising model would have been better (see comments).

Comment 1

Previous studies by the investigators have tested the efficacy of C20 therapy as a daily intraperitoneal injection of 5 mg/kg, for 2 weeks usually (reference 30, 31). The efficacy for T cell activation after C20 therapy was tested in these previous studies with a serum IL-2 response on ConA (reference 31) but not with presence of CD3 cells in blood as in the present study.  Reference 30 mentions a decrease in circulating CD3+ cells after C20 but no difference with control mice not treated with C20, implying a recruitment effect of C20 in the muscle and not a bone marrow or splenic depletion. In the present study, the argument that C20 can be given twice a week and not daily is that the CD3+ (and CD11, but these are leucocytes) blood count decreased in the first two weeks after C20 treatment, not differently between those with daily injection and injections twice a week. This test is however less clear than previous tests (in fact, it has not been compared to IL-2 response), and it may therefore not be excluded that the exercise model has different activations and T cell recruitments necessitating more frequent C20 treatment, than the non exercise model, because we cannot rely on a complete abolition of PCK theta in the exercise model, while we do not know the effects of C20 in the non exercise model. In discussion this remains a speculative reasoning, which would have been less speculative if you could have stated that the C20 inhibition was a complete inhibition or at least that it would be more inhibiting than in the control group not treated with C20. I looked for CD3 myocardial content comparisons in the present paper, presented in Fig 1C: compared with the CD3 content of mdx control mice, the CD3 content of C20 treated exercise mdx  was not much lower, not significantly. What is a concern, is that compared to control mdx mice (not exercising, no C20), there is also no improvement in histology after C20 in those who exercised, so there may not have been a complete inhibition of cardiac abnormality by C20. Maybe the CD3 response/ percentage has been tested further by the authors as a good efficacy measure, and then this should be described. On the other hand, a partial effect of C20 is already a good protection, but then it should be described as such, at least as possibly being a partial response.

Of course, that the effect of C20 is partial is already apparent from the remaining presence of fibrosis and necrosis in the exercising model despite C20, but in how far this is being mediated by T cell infiltration and C20 therapy inhibition, is then not entirely clear. I wonder why you did not use different C20 doses or tried to more specifically see whether this was a T cell problem or not, since you only determined IL-6 mRNA in the cardiac preparations, but not IL-2. Is it still possible to add the IL-2 mRNA results ?

I would propose to better describe the CD3 efficacy measure for C20 therapy, and also describe what argument you find useful to state that C20 therapy improves the original pathophysiology of the (non exercising) mdx model, or that it can be only stated to improve the exercise loaded pathophysiology of the mdx model. For example because C20 has a gradient of responses from 50 to 90% reduction in T cell infiltration CD3+ which corresponds best and is followed by a gradient of responses of 50 to 90 % (so correlating) reductions in fibrosis areas etc. This variable gradient of response may then be due to incomplete inhibition or to an incomplete concept (a partial concept involving T cells).  

Comment 2

Although in the introduction and previous papers of the investigators the link between C20 therapy, PCK theta and T cell infiltration is the pathophysiologic concept, several other hematopoietic cells are seen to decrease after C20, except leukocytes, and certainly, this is an honest presentation of results. How strong is the evidence that T cells are the specific hematopoietic cells that are involved in C20 therapy ? In the abstract you seem to retract from this concept, as no T cells are mentioned anymore, only “immune cells infiltrating the heart”.   

Can you be more precise in discussing the pro’s (present in previous papers in skeletal muscle) and con’s of T cell infiltration as major mechanism ? (Discussion lines 275-282, in line 283-84 you seem to neglect the T cells further, but in previous papers it was possible to time events better and now you don’t know because the hearts were only assessed once at 12 weeks? Does it need further work done ? as the recruitment hypothesis of T cells would need further work). Also state the possible argument for early IL2 asssessment ?

Minor comment

Line 33 : uncurable disease: incurable disease.

Author Response

The manuscript “Inhibition of PKC theta improves dystrophic heart phenotype and function in a novel model of DMD cardiomyopathy” describes the effects of a protein kinase C theta isozyme inhibitor C20 on myocardial inflammation in a Dystrophin myopathy mouse model (mdx), a model previously described in references 43 and subsequently– because of limited cardiac abnormality - adapted with exercise loading in reference 44. The study is interesting and it states interesting ideas about T cell infiltrations in DMD as the initiating pathophysiology, recruited by PKC theta activations. The study is a logical sequel of previous studies on skeletal muscle abnormalities in the DMD model by the same group, and the interpretation of results also rely on these studies.

The authors were faced with a problem of delay in development of cardiac abnormalities in the mouse model, with earliest signs at 12 weeks (reference 44), while also aware of the cardiac effects of a PKC theta knock out mouse model, showing maladaptive cardiac changes (fibrosis and upregulation of ECM and hypertrophy genes) at 8 weeks (reference 53). It means that a PCK theta inhibition cannot start at birth/ too soon when developmental changes are still occurring in the myocardium. Therefore (as illustrated in figure 1) in the present study the mice were treated with C20 or its vehicle from 3 weeks of age with the PKC theta inhibitor C20, the exercise was started at 4 weeks of age and the hearts were studied at 12 weeks. An extra comparison was made with the non-exercise mdx model, which should have some signs as cardiac abnormality as explained in reference 44, but serves as a non-treated non-exercising model. To better understand your findings, a WT control mouse or a C20-treated non-exercising model would have been better (see comments).

  • We thank the reviewer for the comment. However, we need to specify that the mdx model, unexercised, do not show significant cardiac abnormalities at 12 weeks of age. Ventricular fibrosis, infiltration area and heart functionality are comparable to the heathy mice, while cardiomyocyte necrosis is barely detectable. These data led us to use those as controls, since they share the genetic background with the exercised mdx. Given that no cardiac pathology is evident in unexercised mdx mice at this age, as the age-matching WT mice (Morroni et al., 2021), we decided not to treat them with C20. Indeed, we use exercise only to worsen and accelerate dystrophic cardiac phenotype. Lastly, we previously showed that C20 treatment did not alter healthy mice (Marrocco et al., 2017, ref 30). Therefore, also to fulfil the “3Rs” rule in animal experimental procedures, we think that adding WT control mouse or a C20-treated non-exercising model would be redundant and beyond the aim of this study.

Comment 1

Previous studies by the investigators have tested the efficacy of C20 therapy as a daily intraperitoneal injection of 5 mg/kg, for 2 weeks usually (reference 30, 31). The efficacy for T cell activation after C20 therapy was tested in these previous studies with a serum IL-2 response on ConA (reference 31) but not with presence of CD3 cells in blood as in the present study.  Reference 30 mentions a decrease in circulating CD3+ cells after C20 but no difference with control mice not treated with C20, implying a recruitment effect of C20 in the muscle and not a bone marrow or splenic depletion. In the present study, the argument that C20 can be given twice a week and not daily is that the CD3+ (and CD11, but these are leucocytes) blood count decreased in the first two weeks after C20 treatment, not differently between those with daily injection and injections twice a week. This test is however less clear than previous tests (in fact, it has not been compared to IL-2 response), and it may therefore not be excluded that the exercise model has different activations and T cell recruitments necessitating more frequent C20 treatment, than the non exercise model, because we cannot rely on a complete abolition of PCK theta in the exercise model, while we do not know the effects of C20 in the non exercise model. In discussion this remains a speculative reasoning, which would have been less speculative if you could have stated that the C20 inhibition was a complete inhibition or at least that it would be more inhibiting than in the control group not treated with C20. I looked for CD3 myocardial content comparisons in the present paper, presented in Fig 1C: compared with the CD3 content of mdx control mice, the CD3 content of C20 treated exercise mdx  was not much lower, not significantly. What is a concern, is that compared to control mdx mice (not exercising, no C20), there is also no improvement in histology after C20 in those who exercised, so there may not have been a complete inhibition of cardiac abnormality by C20. Maybe the CD3 response/ percentage has been tested further by the authors as a good efficacy measure, and then this should be described. On the other hand, a partial effect of C20 is already a good protection, but then it should be described as such, at least as possibly being a partial response.

Of course, that the effect of C20 is partial is already apparent from the remaining presence of fibrosis and necrosis in the exercising model despite C20, but in how far this is being mediated by T cell infiltration and C20 therapy inhibition, is then not entirely clear. I wonder why you did not use different C20 doses or tried to more specifically see whether this was a T cell problem or not, since you only determined IL-6 mRNA in the cardiac preparations, but not IL-2. Is it still possible to add the IL-2 mRNA results ?

I would propose to better describe the CD3 efficacy measure for C20 therapy, and also describe what argument you find useful to state that C20 therapy improves the original pathophysiology of the (non exercising) mdx model, or that it can be only stated to improve the exercise loaded pathophysiology of the mdx model. For example because C20 has a gradient of responses from 50 to 90% reduction in T cell infiltration CD3+ which corresponds best and is followed by a gradient of responses of 50 to 90 % (so correlating) reductions in fibrosis areas etc. This variable gradient of response may then be due to incomplete inhibition or to an incomplete concept (a partial concept involving T cells).  

We thank the reviewer for the extensive and detailed comment. We addressed all the concerns, as follows:

  • As noted by the Reviewer, we already tested the general efficacy of C20 treatment by IL-2 response on ConA, in our previous work (Reference 30), and we could associate the C20 treatment with a decrease in peripheral blood immune cells (both myeloid and T cells) in the mdx mice. Based on those results, we decided not to repeat the IL2/ConA test, but only to evaluate abundance of circulating immune cells in mdx mice, comparing daily to bi-weekly frequency of C20 administration, also in order to avoid superfluous stress and suffering to the animals.
  • As in any pharmacological approach it is conceivable that PKC theta inhibition in our protocol is not complete, and in any case, it was not our aim to completely inhibit PKC theta. It is also possible that, as the reviewer mentioned, the treatment protocol established in unexercised mdx mice may be less effective when applied to the exercised model, which means that we are underestimating the efficacy. However, our data, using the bi-weekly administration protocol, clearly show that inhibiting this protein, even partially, in an inflammatory environment (as in the ex mdx heart) dampens (not abolishes) overall inflammation and subsequent extensive cardiac damage.
  • Interestingly, we could not find any change in IL-2 in ex mdx (vehicle treated or not) or C20 treated ex mdx at the end of the exercise protocol (when mice are 12-week-old), although we found that CD3+ cells were strongly reduced after C20 treatment.  It is conceivable that T cell activation is an early event, triggering the subsequent late events observed; thus we may have missed the earlier IL-2 peak, since we analyzed the samples at the end of the exercise protocol, when in the ex mdx model the cardiac damage is already established. At present we cannot confirm this speculation, since we did not analyze any other time-points, thus we prefer not to include this data.
  • In conclusion, in this study, we propose inhibition of PKC theta as a possible therapeutic strategy to ameliorate dystrophic cardiomyopathy. We use the exercise protocol as a strategy to worsen and accelerate the pathology not to study the exercise loaded pathophysiology. We thus showed that C20 prevented the worsened pathology induced by the exercise. As in any pharmacological approach it is conceivable that PKC theta inhibition might not be complete as well as that inhibiting PKC-theta dependent mechanisms might not completely resolve cardiac pathology.
  • We modified the discussion to explain these concepts clearly.

Comment 2

Although in the introduction and previous papers of the investigators the link between C20 therapy, PCK theta and T cell infiltration is the pathophysiologic concept, several other hematopoietic cells are seen to decrease after C20, except leukocytes, and certainly, this is an honest presentation of results. How strong is the evidence that T cells are the specific hematopoietic cells that are involved in C20 therapy ? In the abstract you seem to retract from this concept, as no T cells are mentioned anymore, only “immune cells infiltrating the heart”. 

Can you be more precise in discussing the pro’s (present in previous papers in skeletal muscle) and con’s of T cell infiltration as major mechanism ? (Discussion lines 275-282, in line 283-84 you seem to neglect the T cells further, but in previous papers it was possible to time events better and now you don’t know because the hearts were only assessed once at 12 weeks? Does it need further work done ? as the recruitment hypothesis of T cells would need further work). Also state the possible argument for early IL2 asssessment ?

  • Our previous data suggest that PKC-theta is not essential for direct myeloid recruitment to damage site (Lozanoska-Ochser et al., 2018, ref 31), but as literature abundantly suggests, it is for many T cells activities. In line with our hypothesis, the only leukocyte population that was not significantly diminished after C20 treatment is neutrophils, whose recruitment does not require T cells intervention. We thus believe as T lymphocytes inhibition the major mechanism through which C20 exerts its effects, although the eventual contribution of other cell types within the heart cannot be ruled out, as we specified in the Discussion.

Minor comment

Line 33: uncurable disease: incurable disease.

modified

Reviewer 4 Report

The manuscript of Morroni et al investigated an interesting topic, such as, the potential treatment of cardiomyopathy in Duchenne Muscular Dystrophy by targeting the inflammatory component via the inhibition of PKCΘ, expressed by T lymphocytes. 

While the manuscript is well written and the experiments appropriately chosen, described and carefully performed and presented, the main concern is the lack of some control groups.  The authors should show, or discuss with references, similar results obtained from the healthy mice, subjected to exercise in the presence of vehicle or C20, to address the level of exercise induced inflammation in a normal murine heart.

At minimum, the authors should have shown, at least for Fig. 2-4, the results obtained from a different control, consisting of exercised mdx mice untreated with the vehicle. The results for the exercised group treated with vehicle (DMSO) are scattered, and the authors must show that this is not the effect of the vehicle. In this group, the variability is high, nevertheless, in many panels it is possible to depict 2 populations of mdx mice: responding and non responding to exercise. Which ones have been treated with C20?How can the authors be sure that the inflammation is induced by the exercise alone and not by the combination with DMSO? How do they prove that C20 is not treating an effect of DMSO? Therefore, control experiments are missing in order to complete the study.

Minor:

  • the text on the panels: axis titles are too small 
  • in Figure 3, panel D "square mm" should be mm2 for consistency
  • I could not find the age of the mice, please specify.
  • L113- mentions a Supplementary Figure 1 that has not been provided.

Author Response

The manuscript of Morroni et al investigated an interesting topic, such as, the potential treatment of cardiomyopathy in Duchenne Muscular Dystrophy by targeting the inflammatory component via the inhibition of PKCΘ, expressed by T lymphocytes. 

While the manuscript is well written and the experiments appropriately chosen, described and carefully performed and presented, the main concern is the lack of some control groups.  The authors should show, or discuss with references, similar results obtained from the healthy mice, subjected to exercise in the presence of vehicle or C20, to address the level of exercise induced inflammation in a normal murine heart.

  • We thank the reviewer for appreciating our work. As discussed in our reply to Reviewer 1, we previously reported that C20 treatment in WT mice did not alter healthy hearts (Marrocco et al., 2017, ref 31), in terms of heart tissue organization and fibrosis (H&E and Masson’s staining) as well as expression level of specific factors such as ANP, TGF-beta or beta MyHC mRNAs. It is widely reported that moderate “endurance” exercise does not exert any detrimental effect in healthy mice hearts, but is beneficial. In the present study, we use exercise to worsen and accelerate the dystrophic heart phenotype, as a model to test pharmacological approaches. Considering the “3Rs” recommendation in animal experimental procedures, we think that treating exercised healthy mice would be superfluous and beyond the aim of the study.

At minimum, the authors should have shown, at least for Fig. 2-4, the results obtained from a different control, consisting of exercised mdx mice untreated with the vehicle. The results for the exercised group treated with vehicle (DMSO) are scattered, and the authors must show that this is not the effect of the vehicle. In this group, the variability is high, nevertheless, in many panels it is possible to depict 2 populations of mdx mice: responding and non responding to exercise. Which ones have been treated with C20? How can the authors be sure that the inflammation is induced by the exercise alone and not by the combination with DMSO? How do they prove that C20 is not treating an effect of DMSO? Therefore, control experiments are missing in order to complete the study.

  • We previously showed that exercised mdx mice that did not receive either vehicle or C20 treatment, show the same levels of augmented fibrosis and immune infiltration in the heart, as well as reduction in FS, with the same scattered distribution of the values, thus excluding any DMSO effect (Morroni et al., Life, 2021, ref 44). The reviewer correctly noted that some mdx mice respond less than others to the exercise protocol. This is probably due to individual variability, that is commonly high among mdx mice. All the mice were treated with C20, since there is no way to discriminate between “responding” and “non-responding” to exercise in advance. Even if we could, once the mice “responded”, the heart would be already damaged, and our treatment would be superfluous.

Minor:

  • the text on the panels: axis titles are too small 

we increased the size

  • in Figure 3, panel D "square mm" should be mm2 for consistency

modified accordingly

  • I could not find the age of the mice, please specify.

Age is now detailed in M&M

  • L113- mentions a Supplementary Figure 1 that has not been provided.

Supp Figure 1 is now provided

Round 2

Reviewer 1 Report

The authors thoroughly addressed my questions and sufficiently reduced my doubts. Some spelling corrections are still required.  

Author Response

We thank the reviewer for the positive comment. We went throughout the manuscript and check for spelling

Reviewer 2 Report

The authors have made an effort to address the reviewers’ comments. However, there is still weakness in their data interpretation. 

It is the authors’ opinion that the inhibition of PKCθ expressed in T lymphocytes is the major mechanism through which C20 exerts its effects. However, they admit that the eventual contribution of other cell types within the heart cannot be ruled out. In addition, according to the authors, PKCθ is expressed in murine cardiomyocytes but not in the human heart. This fact may require a second thought regarding the ex mdx mouse model as an appropriate model for the present study and raises worries regarding the efficacy of the strategy to inhibit PKCθ as a therapeutic remedy for the treatment of DMD in humans.

Overall, the results in this study are mainly descriptive. The study would benefit from a proposed mechanism (a scheme or discussion) regarding the sequence of the events leading to the reduced dystrophic heart inflammation, necrosis and fibrosis, and improvement of heart function in the ex mdx mouse model under treatment with C20.

Author Response

It is the authors’ opinion that the inhibition of PKCθ expressed in T lymphocytes is the major mechanism through which C20 exerts its effects. However, they admit that the eventual contribution of other cell types within the heart cannot be ruled out. 

We understand the reviewer’s concern about the possible contribution by cell types other than T cells. We addressed these issues more deeply in the discussion.

PKC-theta has been widely described as crucial for the activity of T cells during development and adult life (Refs. 32, 34, 35, 37 + Sun et al., 2000; Manicassamy et al., 2006; Healy et al., 2006; Anderson et al., 2006; Zhorov et al., 2011). Therefore, this protein is considered as a very attractive target for the treatment of inflammatory or autoimmune diseases (Refs. 33, 36, 38, 39 and added refs 67-68). We previously showed, using bone marrow transplantation experiments, that ablating PKC-theta in skeletal muscle but not in hematopoietic cells, did not rescue the dystrophic phenotype in the mdx model (Ref 59). Furthermore, we demonstrated that in skeletal muscle, among the hematopoietic cells, T cells are the main target of PKC-theta inhibition (Ref 30).

Moreover, other anti-inflammatory approaches, not involving the PKC-theta, showed similar results on dystrophic cardiomyopathy in the mdx mouse models (Van Erp e al., 2006; Marques et al., 2009; Zschuntzsch et al., 2020; Laurila 2022, added as Refs 54-57). Taken together, the available data lead us to believe that the anti-inflammatory action of C20 (as a PKC theta inhibitor) might be responsible for the improved phenotype, and that T cells might be the main population responding to C20.

The reviewer might still be concerned about the possibility of a direct and positive C20 effect on murine cardiomyocytes, as they express this protein. While the role of PKC-theta in cardiomyocytes is not clearly elucidated, we previously showed that PKC-theta ablation has a detrimental effect on cardiomyocytes, making them more prone to workload-induced apoptosis (Ref 63). However, PKC-theta is constitutively missing in those mice, while inhibiting the protein by C20 in WT heathy hearts did not adversely affect heart phenotype and function (Ref. 31).

PKCθ is expressed in murine cardiomyocytes but not in the human heart. This fact may require a second thought regarding the ex mdx mouse model as an appropriate model for the present study and raises worries regarding the efficacy of the strategy to inhibit PKCθ as a therapeutic remedy for the treatment of DMD in humans.

Regarding the reviewer’s concern about the fact that PKC-theta is not expressed in human cardiomyocytes, but it is in human T cells, we believe that it is an advantage for the therapeutic point of view, since C20 action would be restricted to the immune compartment.

We modified the discussion to better clarify these issues.

Overall, the results in this study are mainly descriptive. The study would benefit from a proposed mechanism (a scheme or discussion) regarding the sequence of the events leading to the reduced dystrophic heart inflammation, necrosis and fibrosis, and improvement of heart function in the ex mdx mouse model under treatment with C20.

We agree with the reviewer that the study would benefit from a scheme that summarizes the mechanism we propose for C20 action. We added a proposed model as Figure 7. 

Reviewer 4 Report

Probably due to the short time allowed for revision, the authors have chosen to provide a minimalistic response. I agree that the number of mice should be kept low, however, in the cited papers, the number of animals was even lower, therefore they do not account for the variability among individuals, or between generations. Thus, I consider that the main concern about the study has not been addressed, and that the appropriate control is still missing, mainly because the numbers of animals used for both this study as well as for the studies used for references are low.

Although mentioned that the fonts on the graphs have been changed, no improvement could be seen. Also, um is not a measure of length as Fig3C suggests.  Probably a more careful check of the full manuscript would be necessary.

Author Response

Probably due to the short time allowed for revision, the authors have chosen to provide a minimalistic response. I agree that the number of mice should be kept low, however, in the cited papers, the number of animals was even lower, therefore they do not account for the variability among individuals, or between generations. Thus, I consider that the main concern about the study has not been addressed, and that the appropriate control is still missing, mainly because the numbers of animals used for both this study as well as for the studies used for references are low.

We thank the reviewer for raising this issue.

The number of animals used, in our present and previous studies, was kept sufficient to observe statistical differences between the experimental groups, as recommended by the 3Rs rule, taking into account the variance in the distribution of the values inside each group for each different data. For example, WT or mdx mice show a tight distribution of the values of ventricular fibrosis, and the number of 3 or 4 mice was sufficient; for scattered distribution, as for the ventricular fibrosis in ex+vehicle group, even 10 mice were used, in order to account for individual variability. 

We understand the concern about the possible level of inflammation induced by exercise in healthy hearts. Nevertheless, numerous studies have evaluated the effect of chronic, moderate-intensity exercise on the heart of wild-type mice strains and found no detrimental effect on cardiac phenotype and function [1-4]. Instead, beneficial effects of exercise included ameliorated heart function and physiological hypertrophy [5-7], improved antioxidant capacity [8-10], increased angiogenesis [6,11], improved mitochondrial turnover [7,10,12], augmented secretion of cardioprotective factors such as myonectin [13,14], and even telomere-stabilizing protein induction [15].

  1. Feng et al., Life Sci. 2019, 217, 128–140
  2. Perrino et al., Am. J. Physiol. - Hear. Circ. Physiol. 2011, 300, 1983–1989
  3. Ellison et al., Heart, 2012, 98:5-10.
  4. Bover et al., Free Rad. Biol. & Med. 2008, 44, 224 – 229
  5. Kemi, et al., J. Appl. Physiol. 2002, 93, 1301–1309
  6. Gibb et al., Circulation 2017, 136, 2144–2157,
  7. Wang et al., Cell. Physiol. Biochem. 2015, 35, 2159–21684.
  8. Akita et al., Am. J. Physiol. - Hear. Circ. Physiol. 2007, 292,
  9. Yang et al., PLoS One 2014, 9
  10. Musman et al., Free Radic. Biol. Med. 2016, 101, 317–324
  11. Bellafiore et al., Front. Physiol. 2019, 10, 238
  12. Sorriento et al., Front. Physiol. 2021, 12, 660068
  13. Zhou et al., Cell Biosci. 2019, 9
  14. Otaka et al., Circ. Res. 2018, 123, 1326–1338
  15. Werner et al., J. Am. Coll. Cardiol. 2008, 52, 470–482

By contrast, chronic, moderate-intensity exercise in mdx mice increases cardiomyocytes necrosis worsening the dystrophic phenotype; we thus used this protocol to generate a model of dystrophic cardiomyopathy in order to test pharmacological approaches, such as the C20 treatment used in this study.

On top, we previously showed that C20 treatment had no effects on healthy hearts (Ref 31). Considering all these issues, in our opinion exercising WT mice treated or not with C20, would not add further information.

Although mentioned that the fonts on the graphs have been changed, no improvement could be seen. Also, um is not a measure of length as Fig3C suggests.  Probably a more careful check of the full manuscript would be necessary.

We further increased the font size, and changed “um” to  mm. We went throughout the manuscript and check for spèlling.